# An Innovative Approach for the Enhancement of Public Real Estate Assets

**Benedetto Manganelli** [1,*] , **Sabina Tataranna** [1] **, Marco Vona** [1] **and Francesco Paolo Del Giudice** [2]

1   School of Engineering, University of Basilicata, 85100 Potenza, Italy; sabina.tataranna@unibas.it (S.T.); marco.vona@unibas.it (M.V.)
2   Department of Architecture and Design, Sapienza University of Rome, 00185 Rome, Italy; francesco.delgiudice@uniroma1.it
*   Correspondence: benedetto.manganelli@unibas.it

**Abstract:** In a context of dwindling resources and growing financial constraints for public administrations, available real estate assets can become an important economic resource both for debt reduction through their alienation, and for carrying out public works through rehabilitation and defunctionalisation using private capital. The latter requires the adoption of innovative policies and strategies to enhance the value of the assets, especially those that are disused or abandoned, which very often represent a critical element in the overall management of public administrations. This study proposes a strategy for the enhancement of public assets using a little-experimented contractual form of public–private partnership. This approach, through a complex exchange transaction, avoids the total disposal of the assets, thus guaranteeing the social and environmental sustainability of the intervention, and also allows for the enhancement of the property with a financially convenient solution for both partners. The balance between the benefits of the two parties is in fact the basis of the model proposed for the final solution.

**Keywords:** public–private partnerships; real estate; public assets; sustainability evaluation; economic–financial feasibility

## 1. Introduction

Recent and current economic contingencies have forced many public administrations, also under the pressure of European directives, to consolidate their budgets and significantly reduce their expenditure. In this context, public real estate represents an economic resource that can play a decisive role, given that it is very often underused or unused. The management costs of public buildings, related to the operating phase and to ordinary and extraordinary maintenance, are considerable and often exceed the profitability they are able to generate. Moreover, in many cases the functions of administrative activities, without any significant retrofitting, could not be carried out in accordance with the operating and safety criteria as provided by the new generation of regulations, which were created on the basis of technical and scientific developments that are specific to the construction sector.

The optimisation of asset management should therefore be implemented through redevelopment plans or through the divestiture of buildings that are no longer functional for administrative activities.

Property enhancement is often a complex operation which should be carried out with a multidimensional approach that, starting from preliminary feasibility studies and the integrated design of the transformation operations, is able to identify possible alternatives, and selects them on the basis of the needs of the various stakeholders.

In some cases, the development of a solution can be more easily pursued by resorting to the contractual forms of a public–private partnership.

Among the possible forms of partnership to be implemented in public asset valorisation processes, this study has identified one that has been little explored so far, but which

could have a very good chance of success in specific contexts. This is an option provided for in the Italian legislation on public contracts and involves the transfer of goods in exchange for services. This approach, through a complex exchange transaction, avoids the total disposal of the asset, thus guaranteeing the social and environmental sustainability of the intervention, and allows for the enhancement of the property with a financially convenient solution for both partners.

In the following section, the paper analyses the size of the existing public real estate stock in Italy and the various forms of divestment or enhancement usually carried out by Public Administrations, focusing on public–private partnership contracts and their circulation both in Italy and in Europe, as well as the limitations of this instrument as highlighted by the literature on urban regeneration initiatives.

The following paragraphs illustrate a procedure for the enhancement of public property, which is to be implemented through a particular and relatively unexplored form of public–private partnership, through its application to a specific case study. The final paragraphs on results and conclusions show the advantages of applying this form of contract to all the participants involved in the process, on the one hand the public administration which pursues the objectives of environmental, social and economic sustainability but is constrained and limited by strong budgetary constraints, and on the other, the private investor who tries to optimise profits while reducing risks as much as possible. The case study presented here is proposed as a best practice for other public real estate redevelopment projects.

## 2. The Management and Enhancement of a Public Real Estate Asset

The financial crisis and the state of indebtedness of many public administrations have changed the strategies for the management and enhancement of the value of public real estate assets [1]. From a hitherto conservation-oriented property management implemented to date, which we could define as static, there has been a shift towards a dynamic form of management in which assets become an opportunity and therefore a source of income [2]. The change in strategy was also determined by the not negligible percentage of unused properties, which in Italy reaches 7% of all public assets. It should also be noted that a large part of these assets is in such a state of obsolescence so as to require maintenance or renovation whose costs are not sustainable for the budgets of the public administration [3].

Spending rationalisation has often pushed administrations to seek resources through plans to alienate public assets. In other cases, given the very low return on assets even when used for non-institutional purposes (in 2011 the return reached peaks of 1 and 2 per cent per year before expenses, taxes and management costs [4]), innovative management tools, such as global services, project finance and property leasing, have been used.

The choice between divestment and enhancement is conditioned by several factors. On the one hand, there are economic, political and social reasons that might lead to the preservation of the property rather than its divestiture. On the other hand, the property's current ability to generate income is an important consideration [5]. If profitability is already positive, and the political, economic and social reasons for the transfer of the property subsists, divestment will be a relatively simple operation. When there is no profitability or if it is negative, then both preservation and sale go through a preliminary enhancement operation.

The process of enhancement must first of all verify the sustainability of the preservation of the building, or its demolition and reconstruction [6]. The state of abandonment of an individual building or groups thereof, is a condition dynamically intertwined with the process of decay, which is often generated by external forces, be they demographic, economic or social, but which in turn produces negative externalities in terms of perceived quality, attractiveness and property values of the entire surrounding area as well [7–10]. However, due to such dynamic intertwining, abandoned properties can offer opportunities to reverse degradation processes and become part of an urban regeneration strategy [11]. It must be stressed that on an urban scale, the advantage of preventive intervention has been

widely acknowledged and quantitatively assessed in terms of resilience [12], while the lack of suitable and widespread prevention strategies (maintenance interventions) have shown mostly devastating effects in the past decades [13].

Currently, one of the priority objectives of European government policies is to increase the sustainability of the built environment in economic, environmental and social terms [14]. Government incentives for urban regeneration processes [15] are the tool of choice to foster sustainable development and smart growth in cities. Famous European examples, such as the Berlin building recycling model [16], the building redevelopment of Hamburg port, the Amsterdam residential redevelopment [17], or the Brownfields redevelopment [18], etc., show that urban regeneration represents a true opportunity for the smart growth of cities.

The international slogan of the 3R environmental movement—Reduce–Reuse–Recycle [16], which incorporates the so-called waste hierarchy, has been extended to architecture in order to create a possible hierarchy of change strategies in the management of the existing building stock. In general, the reduction of consumption in the building process results in a minimisation of interventions and demolitions. The accurate perception of the building's condition, the tenants' conduct and the correct planning and implementation of the maintenance interventions appear to be crucial in preventing demolition. However, the reuse of the existing building stock can only come about by following in-depth renovation works that also differ according to the state of use of the building and its structural and functional suitability in relation to its specific destination. These problems are particularly relevant for reinforced concrete buildings, which represent a large share of the building stock. They require significant energy and structural and functional interventions.

Energy retrofitting is consistent with the objectives of achieving climate neutrality and reducing emissions, which are closely linked to the new standards proposed on 14 December 2021 by the EU Commission in the updated Energy Performance of Buildings Directive (https://energy.ec.europa.eu/topics/energy-efficiency/energy-efficient-buildings/energy-performance-buildings-directive_it, accessed on 10 January 2022). Failure to comply with the measures could lead to a ban on the sale or rental of buildings.

Regarding structural rehabilitation, it should be noted that it has not received the same level of importance and attention from the European government. However, the whole Mediterranean area and the Italian territory are affected by a significant seismicity which, combined with the high level of seismic vulnerability, has implied serious consequences even for seismic events of medium intensity.

Moreover, the Italian reinforced concrete buildings are among the oldest in Europe. Construction data show that many buildings were constructed at a time when no seismic action was foreseen, or the design methodologies were based on very poor standards that did not consider the necessary durability and conservation requirements.

Lastly, regarding functional upgrading, in those buildings still in use modernisation is aimed at achieving an upgrade to the current standards. On the other hand, for buildings that have fallen into disuse and are in a state of total abandonment or for those that are still in use but close to abandonment, renovation is understood as a process of revitalisation. This process can take the form of transformation, re-design of spaces, modification of the original structural layout by increasing or reducing existing volumes and creation of new spaces within and between existing volumes [19]. All the above-mentioned renovation strategies are not to be understood as mutually exclusive, but can be implemented simultaneously, depending on what the design objectives and the demands of the customers are. In these strategies, recycling is conceived not only as the recycling of building materials but also as the recycling of design, form and building typologies, which can, in turn, be transposed to the current context in a modern manner.

For private properties, when choosing the redevelopment strategy, the specific interests of the owners play a crucial role [20,21]. As an example, demolition can mean 'destroying the initial capital, not only in economic terms but also in terms of social sustainability, cancelling the valuable interpersonal relationships that have developed between residents

over the years' (https://www.world-architects.com/it/projects/view/rotkreuz-highlife, last accessed on 10 May 2021).

Implementing virtuous strategies on public real estate assets is an equally complex task [22].

Such operations require not only a technical feasibility study aimed at selecting the optimal intervention strategy [23,24], but also an accurate study of the economic–financial feasibility of the project.

The difficulties resulting from the economic crisis that have characterised the recent decades have often made public–private partnerships (PPPs) the only viable contractual form for financing urban regeneration projects or public assets [25–28]. Despite this awareness, this contractual form for the realisation of public projects is still little used. The reason for this is often the inability of public authorities to independently develop the preliminary stages of technical, legal and economic–financial feasibility analyses [29]. Many initiatives at the European level have tried to foster public–private partnerships. One such initiative, implemented by the European Commission with the support of the European Investment Bank, is JESSICA, which aims to help Member State authorities exploit financial engineering mechanisms to support investments in sustainable urban development.

In Italy, to overcome these obstacles, with the entry into force of Law 28 December 2015, no. 208, (art. 1 paragraph 589), the Department for Economic Policy Programming and Coordination, (DIPE) has the task of aiding and supporting all central and local Public Administrations, who request this, in all those phases related to PPP operations, from planning to implementation.

Where the use of PPPs to finance urban regeneration projects is more common, this approach suffers from a number of deficiencies. These include the project's vulnerability to financial risks due to over-reliance on private investment, and the overemphasis on creating a place of opportunity rather than improving the social environment in disadvantaged areas [30]. Some authors show the limitations of this tool when it is considered the universally applicable solution to urban regeneration problems [31]. For the success of an urban regeneration initiative through private contribution, the use of strategies adapted to specific contexts is therefore crucial [32,33].

With the exception of the United Kingdom, the PPPs market in the EU, besides the crisis that has also affected this specific type of activity on a European scale and that has led to a constant decline in the last decade, still has ample margins for development (Figure 1).

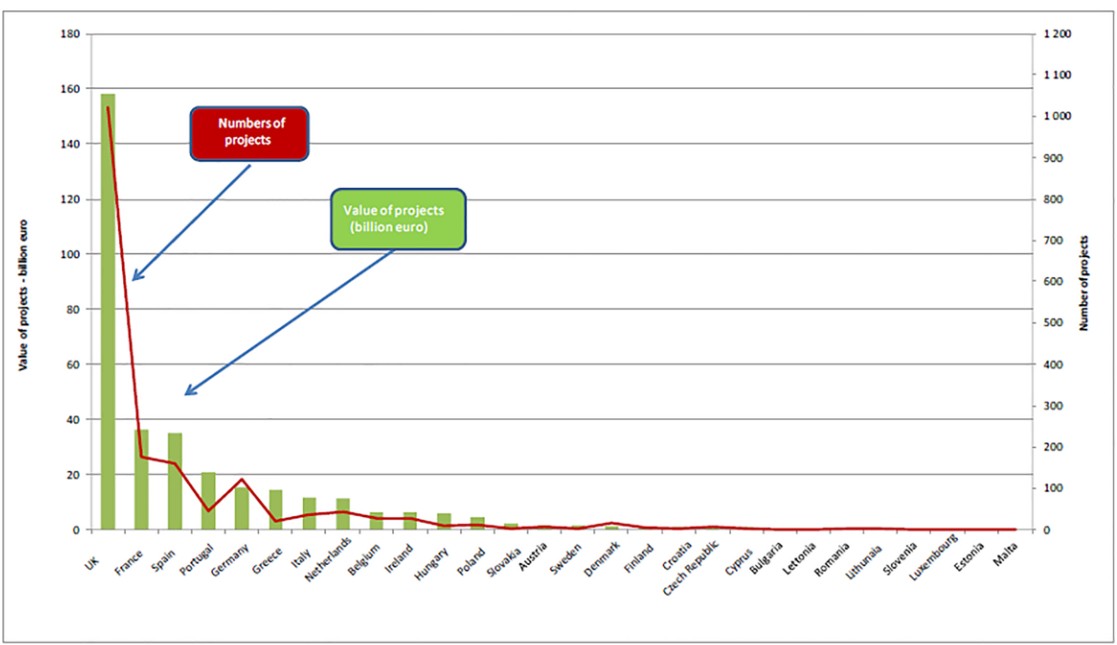

**Figure 1.** PPPs market in European countries (Source: European Court of Auditors, Special Report, 2018).

In Italy, also due to the low circulation of these contractual forms in comparison with other European countries, in the last 18 years, from 2002 to 2019, there has been a relative increase in the number of initiated proceedings (approximately 40,000 have been initiated) as well as tenders awarded and in progress, for a total amount of more than €100 billion (Figure 2).

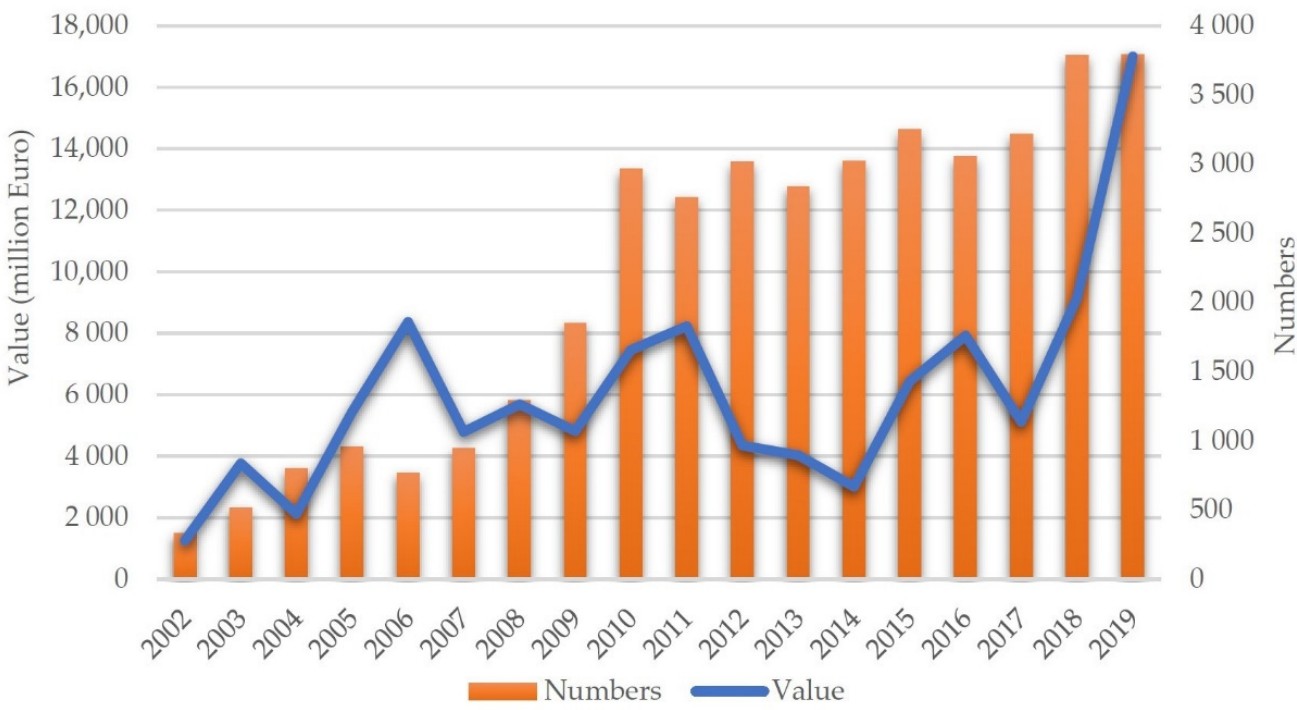

**Figure 2.** Italian PPPs market from 2002 to 2019. Value and number of calls for tenders per year (Source: DIPE processing of data from the National Observatory of Public Private Partnerships).

### 3. Enhancement Strategy

In Italy, the legislative framework for public procurement is regulated by the Public Contracts Code (Legislative Decree 18 April 2016, n. 50).

Among the possible forms of PPPs envisaged by the Code there is the Transfer of assets in exchange for works (art. 19). To date, this contractual form has rarely been applied in cases of alienation of assets of local Public Administrations and, so far, in no case has it been used for alienations of assets of the central government.

This form of partnership is the essential element of the property enhancement strategy proposed in this paper. The use of this form of partnership is particularly appropriate in the process of increasing the value of public property assets, carried out in conjunction with rationalisation plans aimed at the partial transfer of assets, because of the advantages it can bring to both partners involved. The private partner finds an advantage, in the first place, in the reduction of the initial investment cost, understood as the capital outlay needed to acquire the asset that is to be redeveloped. The reduction of the initial investment cost implies a consequent reduction of the financial risk.

The possibility provided by the regulation that entails disposal of a property before the end of the works upon presentation of a suitable bank guarantee for a value equal to that of the property to be acquired, also reduces the degree of financial illiquidity. The public partner, on the other hand, benefits from the possibility of being able to undertake virtuous redevelopment operations, also aimed at functional conversion, without using financial resources but making use of the assets themselves as the only resource.

Based on these prerogatives, the proposed enhancement model was developed.

The proposed property enhancement strategy derives from the construction of a valuation process developed in steps according to the hierarchical order of Figure 3.

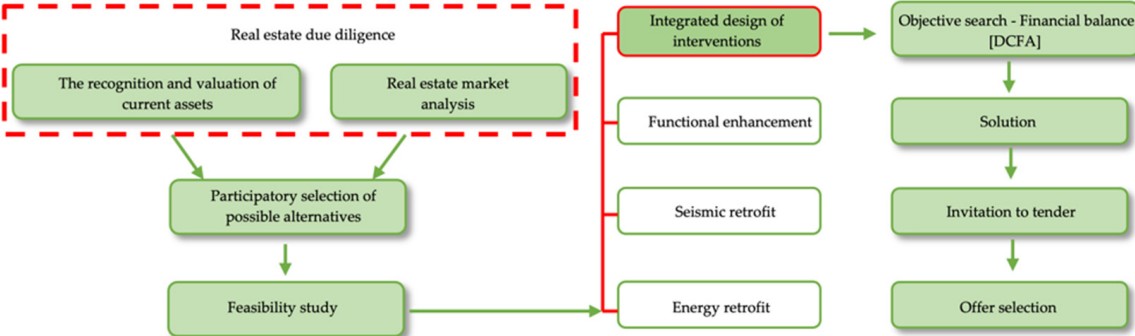

**Figure 3.** Stages of the enhancement strategy.

Based on the information collected during the initial cognitive phase, the different possible transformation/development alternatives should be identified. The specific needs of stakeholders (direct and indirect) are investigated through interviews aimed at guiding the decision-making and design process from a participatory planning perspective [34]. In this way, the social impact is incorporated into the technical and economic feasibility project, leading to the selection of the best option as a balance between the different interests of the stakeholders involved in the transformation process. Once the technically and socially sustainable transformation hypotheses have been identified, the final choice is selected as the outcome of financial sustainability from the private partner's point of view.

The financial sustainability of the operation is developed through the Discounted Cash Flow Analysis. The use of this methodology implies, on the one hand, the estimation of the costs of the restructuring operation and the management of the PPPs contract and on the other, the forecast of the probable revenues. The estimation of the technical costs can be preceded by the creation of a BIM model of the building, whereas revenues are derived from the analysis of the market and the potential demand in the scenario subsequent to the transformation process. The development of the DCFA thus allows to define the margins of the financial and economic balance between the public and private sectors, and therefore the terms of the PPPs contract for the formulation of the tender.

As a matter of fact, the main element of the contract is in fact the portion of the building that is to be transferred to the private partner who will undertake the renovation of the whole complex. This parameter does not result from the equivalence of the value of the part to be sold, with reference to the current conditions of use and maintenance, and the overall cost of the renovation. Instead, it is the result of an analysis that ensures the financial sustainability of the private investment. The result is therefore determined by a DCFA that has as an unknown the extent of the surfaces that will be transferred to the private sector, or, as a difference, the extent of those that will remain at the disposal of the public sector, once all the other parameters are known. These include transformation and management costs, revenues and the minimum profit expected by the ordinary entrepreneur. Only this approach can guarantee the success of the operation and the triggering of a virtuous process of redevelopment and regeneration.

The private partner will have to bear all the investment costs related to the renovation and redevelopment of the building. To this, one must add the cost of the bank guarantee necessary to obtain the availability of the part of the building defined at the end of the tender, at a time prior to the completion of the works. This will make it possible to plan the sale of the areas to be redeveloped already during the execution of the works, thus reducing the financial exposure. The revenues of the operation are derived from the sale or income of this portion of the property. Furthermore, of fundamental importance, is the condition of lower financial risk to which the investor is exposed due to the lower investment cost.

In construction it is common for investors, to reduce their financial exposure, to require an exchange of real estate. Instead of buying the development land, they transfer to its owner a share of the building to be constructed. Similarly, also in this case,

part of the financial and market risk is thus transferred from the investor to the owner (the public administration).

The public partner, in view of the loss of part of the assets, or rather the lower income that derives from the sale of the entire building in its current condition (of abandonment and degradation), will obtain the restitution of the part not transferred to the private partner but upgraded, which can thus possibly be sold or rented out or used for institutional purposes.

The public partner, from an economic point of view, will also benefit from the redevelopment of the interior of the building which, as mentioned, could become an opportunity to reverse the processes of degradation that characterise the area. This way of enhancing the value of the building is perfectly oriented towards the achievement of sustainable objectives in the management of the existing public assets: reduction of consumption (deriving from the elimination of the management and maintenance costs of disused buildings, as well as from the energy requalification of the part that will remain in its possession), re-use through the technical–functional upgrade of the spaces and recycling through the functional conversion adapted to the demand.

The amount to be transferred to the private partner is the result of objective research able to guarantee the financial equilibrium for the private investor. The only constraint placed on the objective research is that the final value of the part that will remain in public ownership exceeds any proceeds from the sale of the entire property in its current condition. The result of the financial balance defines the basis for the tender.

The contract can be assigned through the Most Economically Advantageous Tender (MEAT), requesting offers from potential participants that meet both quantitative (a reduction or an increase in the size of the area to be transferred to the private partner) and qualitative (regarding materials and workmanship) criteria.

## 4. The Case Study

The case study is a building complex located in the city of Potenza (Southern Italy), near the historic centre. The city of Potenza is the capital of the Basilicata Region, as well as the most populous municipality in the entire region with a population of over 67,000 inhabitants and a demographic weight of approximately 12% on a regional basis and 18% on a provincial basis. The population density of 382.6 inhabitants/km$^2$ is one of the lowest among the 144 Italian cities with a population of more than 50,000 inhabitants. The building complex consists of three buildings with a reinforced concrete frame structure, built adjacent to each other but structurally independent. Its location is particularly advantageous due to its proximity to the main historical and cultural sites of the city (Figure 4), and its easy connection to the main roads and basic services of the city. The urban fabric in which the property is located is predominantly residential, with the presence of various types of commercial and service activities.

In this area, the building and urban planning interventions allowed are aimed at the qualitative maintenance of the existing building stock, at the qualification and greater provision of equipped public spaces and at the encouragement of the presence of activities compatible with and complementary to the residence. The retrofitting intervention provided for is perfectly coherent with the building renovation categories of the urban context and perfectly in line with the objectives of the recent regulations in terms of energy and structural upgrading.

One of the features that makes the case study interesting is the ownership structure and the different uses coexisting therein. One part of the building is used for private residences, the other part, in disuse, is publicly owned and is intended for school and offices. The internal spaces of this portion are therefore functional for teaching activities and distributed in classrooms, reading rooms, libraries, offices and services.

These spaces, unused for about twenty years, are currently in a state of abandonment and progressive degradation. The deterioration is also an indirect consequence of the PA's lack of financial resources, which has led to a low frequency of ordinary and extraordinary maintenance activities.

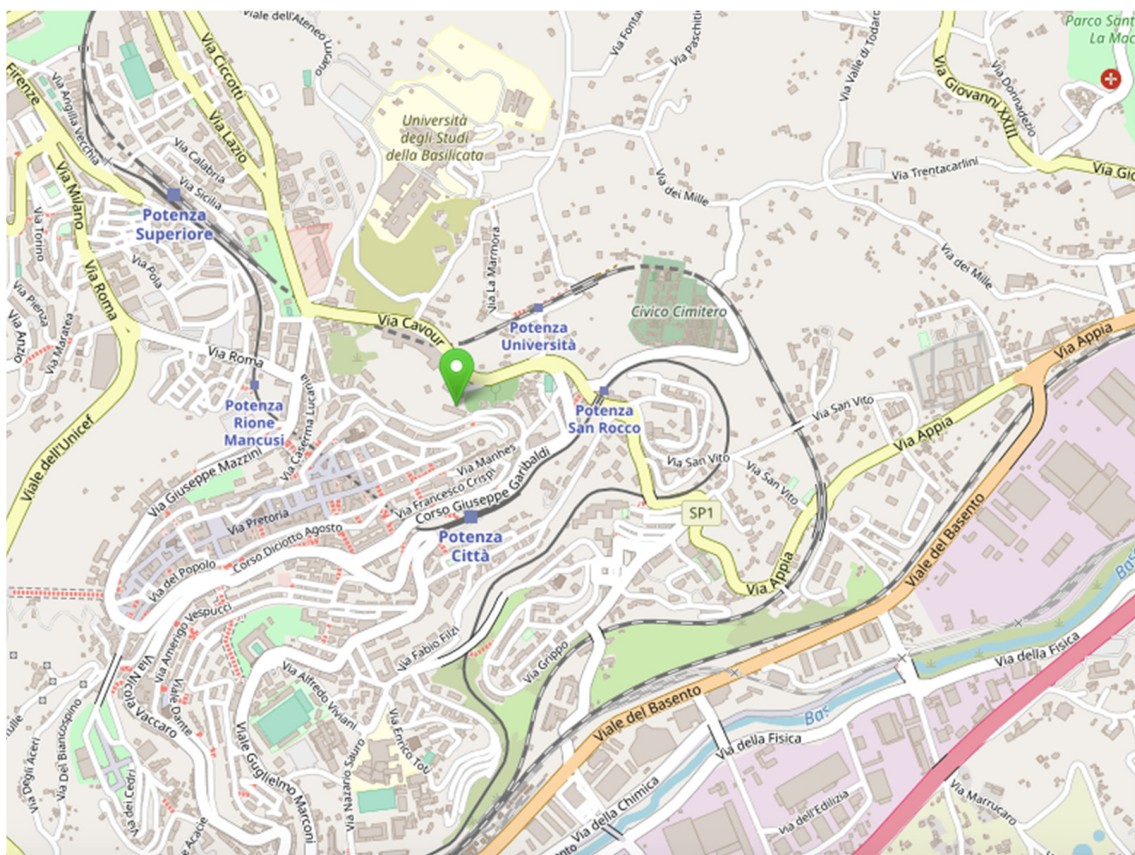

**Figure 4.** Location of the building within the city (Source: OpenStreetMap).

The building complex is now an instrumental asset that is no longer functional for the activities of the public authority. This condition has led the administration to include the asset in the list of possessions available for sale, or alternatively to look for a solution that would enhance its value without depleting its own resources.

From a structural point of view, the building complex was constructed between the late 1960s and early 1970s. It was designed in accordance with generally obsolete regulations and lacking seismic classification. The three buildings (named A, B, C) are arranged on a natural slope and for this reason they have separate foundations on different levels (Figure 5). The three buildings have 11, 7 and 9 floors respectively. A and C have in addition a sloping roof. The covered surfaces of the typical floor layout are respectively about 280 m$^2$ (building A), 180 m$^2$ (building B) and 280 m$^2$ (building C).

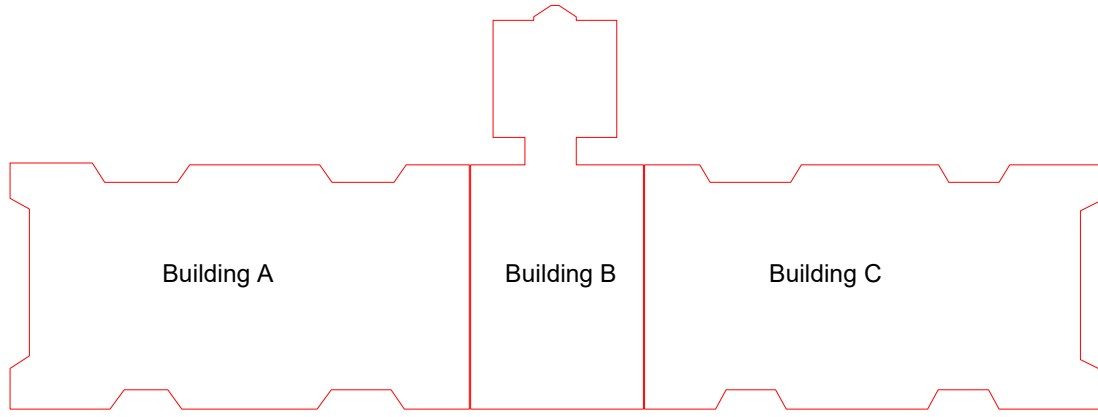

**Figure 5.** Plan showing three buildings.

The total surface area of the building complex is 5885 m², of which 3450 m² are owned by the Public Authority, as shown in Figure 6.

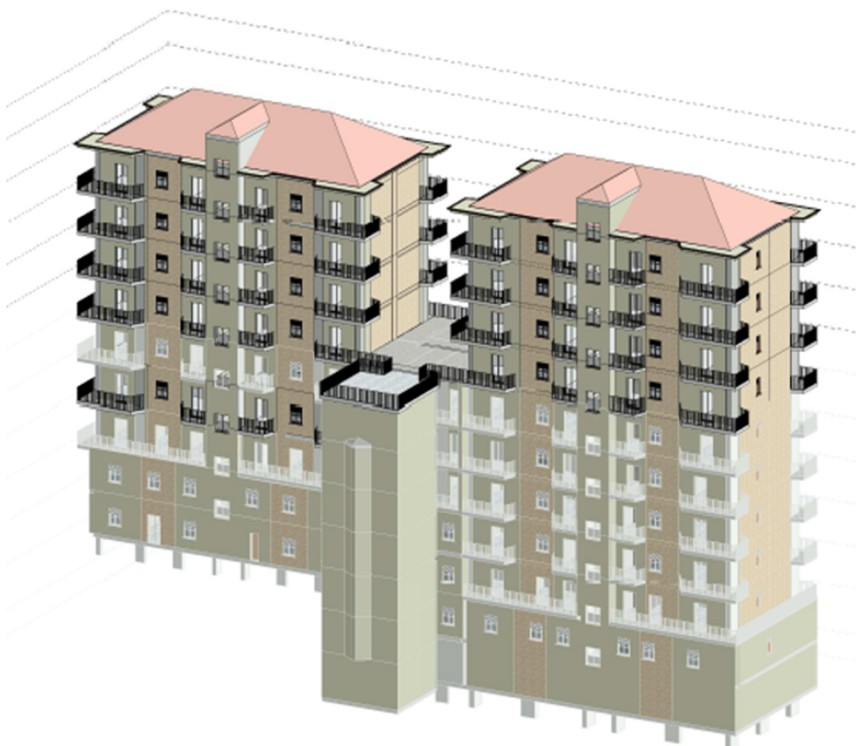

**Figure 6.** Rendering of the case study. The dark part indicates private property, the light part public property.

At the time the assessment was carried out, no original structural design drawings were available. In order to assess the safety level and to define the intervention strategy, an accurate geometric survey of the structure was carried out to compensate for the lack of information. In its initial phase, for the acquisition of knowledge of the reinforcement details, a simulated project of the buildings was created, corrected and integrated with the information found through in situ surveys.

Over the years, no structural works have ever been carried out, and considering the current regulatory requirements and the current seismic classification, a seismic improvement/adaptation was deemed necessary, without which the use for public purposes (such as student services) would not be appropriate. Moreover, the lack of energy efficiency measures would make its management extremely costly.

*4.1. Revenue Estimation*

Market analysis is also part of real estate due diligence. From this, possible scenarios can be deduced in terms of market values and expected revenues from sales resulting from the transformation, rationally orienting project choices, such as intended use, building quality, etc. The real estate market framework was carried out based on information extracted from the 2020 Real Estate Market Observatory Reports (Source: Revenue Agency).

The transformation project has been defined to achieve a real estate enhancement, therefore an increase in the probable expected market value. The internal redevelopment work will shift the existing residential units and the additional ones that will be created with the planned intervention, from the initial "low-cost/ inexpensive dwellings" classification, to the "high-cost/expensive dwellings" type. While the first type currently has average prices of around €900/m², the prices of the transformed product fluctuate between €2000/m² and €2300/m². The hypothesised regeneration and redevelopment

scenario has used the maximum value of this range as the probable sales value of the new residential surfaces.

The hypothesis that the envisaged intervention can generate that surplus on the final values is also confirmed by previous studies on the effects of energy retrofitting on the Italian urban real estate market [35,36].

The enhancement project foresees, for the publicly owned surfaces (including those that will be transferred to the private partner), in addition to energy retrofitting, structural retrofitting, extraordinary maintenance and defunctionalisation interventions.

The new internal layout is oriented towards the subdivision of the different floors into flats. The segment chosen is that of flats with surfaces varying from a minimum of 70.9 m$^2$ to a maximum of 105 m$^2$ (Figure 7). The flats with larger surface areas are those located in the central building, where the position of the external entrances does not make it possible to reduce the size of the apartments.

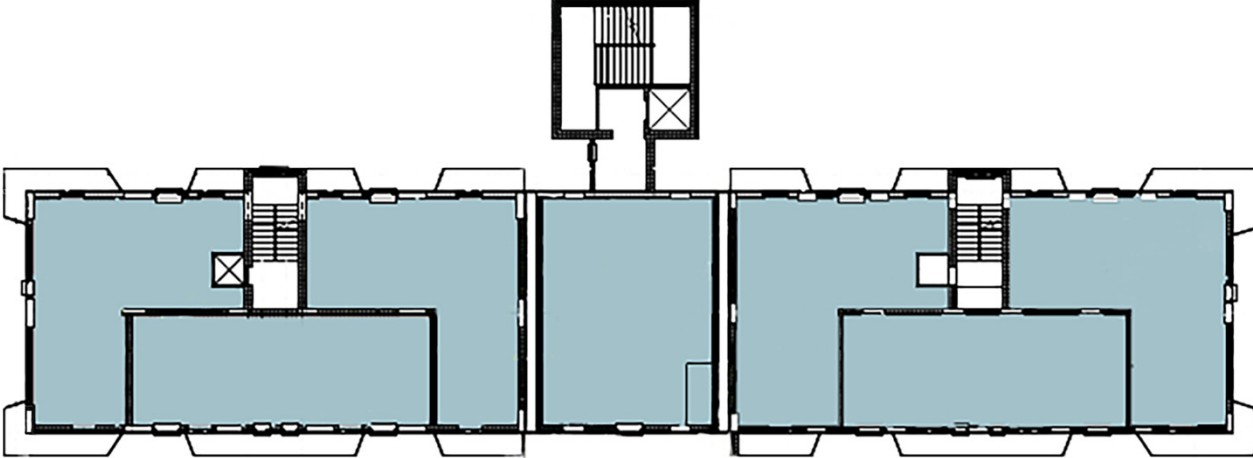

**Figure 7.** Articulation of the standard floor in the new flats.

The identification of the size class derives from analyses of the local real estate market, which showed that in 2020 the most traded size class, by far, was between 50 and 85 m$^2$, followed by the size class between 85 and 115 m$^2$.

The ground floor has two underground sidewalls and will be used for storage rooms/ garages belonging to the flats. The areas of the first, second and part of the third floors, given their exposure and luminosity, will be used as guest quarters and represent the minimum portion that, following the conversion, will remain available to the public body. Part of the third, fourth, fifth and sixth floors, on the other hand, are susceptible to both conversions (guest quarters or flats). The portion of the surface area to be used for flats, which will be transferred to the private partner, and, by contrast, the part to be used for guest quarters, which will remain at the disposal of the public partner, represent the unknowns of the problem, the solution to which derives from the subsequent assessments of the economic–financial sustainability of the conversion.

### 4.2. Estimation of the Transformation Costs

For this case, several retrofit strategies have been taken into consideration, critically analysed and qualitatively compared in order to select the best intervention solution in terms of structural safety, costs, architectural and functional requirements. Of the need to intervene without interrupting the use of the structures in the residential portions was considered of particular importance. The retrofit strategy was selected due to: (1) the need to minimise the impact on the original architectural and functional characteristics of the buildings; (2) the safety requirements of the seismic regulations; and (3) the initial results of the acquisition of knowledge process of the structure.

The selected structural intervention has been operationally defined, consistent with the technical–economic feasibility study of this present study. It provides for the introduction of passive protection devices below the second deck of buildings A and B and then at the base of building C, in order to allow for the correct implementation of a seismic isolation system at the base of the whole complex under examination and in respect of the required displacements. The first two rows of pillars of buildings A and B will have to be reinforced accordingly.

As far as the economic aspect is concerned, in cases similar to the case study, the seismic isolation system has already shown its convenience compared to other systems [24,37].

The insulation system was designed, essentially for cost analysis purposes, considering elastomeric and sliding insulators designed according to the requirements of the Technical Standards in force. For the stiffening at the base of the superstructure, a grid of steel beams was considered, also useful for the connection between the insulators and the superstructure.

The energy upgrade project includes the efficiency enhancement of the external covering of the entire building complex. In particular, the project provides for the application, on the opaque vertical structures, of an insulation system from the outside, comprising expanded polyurethane panels (EPS), the replacement of the external window frames with new ones, and the construction of an insulation system at roof level through a system of insulation from the outside in extruded polyurethane panels (XPS). The adaptation of the plant engineering and, in particular, of the heating systems was designed and calculated only for publicly owned areas.

For these areas, the internal renovation and extraordinary maintenance works involve the redefinition of the internal layout through an almost total demolition of the internal partitions and the construction of new finishes and coverings on the floors and walls. The plumbing and electrical systems will also be upgraded.

The following are also planned: the renovation of the facades not affected by the insulation system, the renovation of the screeds and finishes of the balconies, the replacement of the railings, the replacement of the downpipes and drains and the renovation of the floor of the terrace that cover the central building.

The assessment of the costs of materials and works was developed by means of the bill of quantities on the basis of the three-dimensional BIM model of the building and official regional price lists.

The estimate of the works led to a total construction cost of €3,531,557.88. The cost of the works related to the energy upgrade of the buildings has an incidence on the total cost of 18%, while the cost of the seismic retrofit is higher, at about 31%. The cost of facade upgrading has an incidence of about 9%. The remaining amounts are related to the internal renovation works for functional conversion.

These costs have been appropriately allocated to the different building units, on the basis of their ownership thousandths shares. The estimated average expenditure of each co-owner (private ownership of the building complex) is €9,987 for energy upgrade interventions; €17,070 for seismic retrofit interventions.

The expenses for the seismic retrofit, energy upgrade and facade rehabilitation can be financed for the private part through the tax incentives currently available. These incentives make it easy for these condominiums to take part in the intervention, even if the public body as the owner of the relative majority of the total property did not have any bureaucratic obstacles. The private part of the building complex obviously does not enter into the financial analysis, which focuses only on the public portion from which forms of financing through tax incentives are excluded. As a matter of fact, the current legislation does not include the public subject among the beneficiaries of the incentives.

The final cost of the transformation process is obtained by adding the technical cost deriving from the bill of quantities to:

- The technical and management expenses incurred by the processing company and estimated as a percentage of the construction cost;

- The marketing expenses calculated as a percentage of the total revenues from sales;
- The Urbanization Charges and the Construction Cost Contribution calculated on the basis of parameters provided by the municipal administration;
- The interest charges calculated as a percentage of the investment cost;
- The cost of the bank surety policy.

## 5. DCFA Results

The investment period is assumed to be 28 months discretised into four-monthly intervals. The total cost of production has been broken down into different shares in relation to the different types of ownership and the specific form of financing of the intervention. The surface areas to be transferred as compensation for the works carried out, are those that are able to compensate the costs for the conversion of the surface areas incurred by the investor and guarantee an adequate profit. The cost flow in the different four-monthly intervals derives from the preliminary breakdown of the project into distinct phases that represent the activities that are necessary for the progress of the project, and by taking into account any interdependence between them. All the costs related to the seismic retrofit and energy upgrade are incurred in the first year. The works related to the renovation and transformation of the interior surfaces are distributed throughout the entire period of the investment. Revenues from the sale of the flats are assumed to be distributed over the entire investment period.

The sale before completion of the works is possible as a result of the transfer of the portion of the property to the private partner, subject to the signing of a surety policy contract. The estimate of the surface area to be transferred to the private partner derives from an objective research procedure, assuming this parameter as the unknown of the DCFA, which has a discount rate equivalent to the return expected by the investor. The only constraint placed on the objective research is that the final value of the part that will remain at the disposal of the public body exceeds any proceeds that derive from the sale in the current condition of the entire property. Assuming that the expected profit is equal to 20% of the investment cost, the surfaces to be transferred to the private partner are 1143 $m^2$, which correspond to about 30% of the total surfaces available. This share is lower than the total amount of transferable surfaces (1820 $m^2$), i.e., those that can be converted into residences because they meet the health, hygiene and habitability requirements of the building regulations in force. With regards the public administration, the financial convenience of this solution with respect to the total sale of the property also derives from the verification that the final value of the property, that will remain available to the local authority, is greater than the value obtainable from the sale of the entire property in its current condition.

## 6. Conclusions

The concrete need to find a financially sustainable and economically, socially and environmentally viable solution for the enhancement of public real estate assets otherwise destined for sale, was the basis for this study. The solution found is the use of a form of public–private partnership envisaged by Italian law but which has not been applied in practice so far. This form of contract involves the participation of a private partner as the financier and executor of the process of transforming and enhancing the value of the property, in return for the transfer of a portion of the property. The application to a concrete case has shown that the sale of the asset, as an alternative choice to the use of this contractual form, would produce lower revenues than the utility deriving from the availability of a portion of the same building but redeveloped at the expense of the private partner and destined to a new public function.

In addition to the financial benefits, there are also the positive effects of urban regeneration that the redevelopment of a disused public asset can generate in a wider urban area, and by simply being the trigger for a virtuous process of development and new investments. On the other hand, the private partner would have the possibility to reduce the financial

exposure of the investment and thus reduce the related risks. This condition makes the investment attractive to a wider range of private actors. The increase in supply can only generate further advantages for the public administration. These advantages can be targeted on specific objectives through the selection of the private partner via a Most Economically Advantageous Tender. The selection of the private partner will therefore be carried out on the basis of qualitative and quantitative criteria and through the weighting thereof in line with the objectives of the development. The case study demonstrates the advantages of using this form of contract and also provides a guide to the choice of this solution for administrations with disued real estate assets or those already included in divestment programs, but where the alienation procedure has failed and where they are therefore looking for a solution that can enhance their value without using their own resources.

**Author Contributions:** Conceptualization, B.M., S.T., M.V. and F.P.D.G.; methodology, B.M.; validation, B.M., S.T., M.V. and F.P.D.G.; formal analysis, investigation and data curation, B.M. and S.T.; writing—original draft preparation, writing—review and editing, B.M., S.T., M.V. and F.P.D.G. All authors have read and agreed to the published version of the manuscript.

**Funding:** This research received no external funding.

**Institutional Review Board Statement:** Not applicable.

**Informed Consent Statement:** Not applicable.

**Data Availability Statement:** Not applicable.

**Conflicts of Interest:** The authors declare no conflict of interest.

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
