# Peer review of "An Innovative Approach for the Enhancement of Public Real Estate Assets"

_sustainability, doi:10.3390/su14148309_

Round 1
Reviewer 1 Report
Please see the attached file.

Author Response
Dear Editors and Reviewers,
We quite appreciate the reviewer’s insightful comments. Now we have revised the manuscript.
The literature on urban regeneration was examined in depth and the organization of the paper (R3) was described at the end of the introduction.
References have been checked (R2).
We make it clear that the originality of the proposed approach does not concern public-private partnership in general, which as better illustrated in this corrected version of the manuscript, albeit with many limitations, is widely applied in the processes of urban regeneration and enhancement of public assets, but a specific form of PPPS (R1).
We did not take up the first reviewer's suggestion asking for more detail in the financial analysis. Any changes would, in our opinion, have made the paper too close to a case study, also in contrast to the second reviewer's suggestion (which instead criticises the overemphasis on the financial aspect). By highlighting the positive aspects of the specific form of partnership proposed, the paper is intended as a general guide and best practice.
Reviewer 2 Report
Well written paper in an important area with focus on Italy as the case example. Not sure I agree this is always necessary: "the final choice is selected as the outcome of financial sustainability..." and I would argue that in fact much more emphasis ought to be put on environmental and social sustainability instead of such focus on the economic model prevailing as the dominant force for decision-making. Checking of referencing is required.
Author Response

(The authors gave the same response as above.)

Reviewer 3 Report
The paper is properly structured and addresses a relevant topic. However, in order to acquire an interest for the scientific community, it would be necessary to deepen the literature on urban regeneration and the redevelopment of a disused public asset, from an academic point of view. Furthermore, a methodological section is needed in the introduction as well as the description of the structure of the paper. The conclusion should be enriched with a reflection of a possible replication of the lessons learned in an international context, if not, the risk is for the paper to have a very limited scope and interest.
Author Response

(The authors gave the same response as above.)

Round 2
Reviewer 1 Report
The authors' stance for the paper is that the manuscript is subject to a general guide and best practice. I might give this paper the "possible-to-be-published" status as my point of view is opposite to the other reviewer.
Reviewer 3 Report
I consider the paper adequately revised to be published